# Pyrosequencing Analysis of O-6-Methylguanine-DNA Methyltransferase Methylation at Different Cut-Offs of Positivity Associated with Treatment Response and Disease-Specific Survival in Isocitrate Dehydrogenase-Wildtype Grade 4 Glioblastoma

**DOI:** 10.3390/ijms25010612

**Published:** 2024-01-03

**Authors:** Fábio França Vieira e Silva, Marina Di Domenico, Vito Carlo Alberto Caponio, Mario Pérez-Sayáns, Gisela Cristina Vianna Camolesi, Laura Isabel Rojo-Álvarez, Andrea Ballini, Abel García-García, María Elena Padín-Iruegas, Jose Manuel Suaréz-Peñaranda

**Affiliations:** 1Department of Medicine and Dentistry, University of Santiago de Compostela, San Francisco Street, s/n, 15782 Santiago de Compostela, Spainmario.perez@usc.es (M.P.-S.); giselacristina.vianna@rai.usc.es (G.C.V.C.); abel.garcia@usc.es (A.G.-G.); jm.suarez.penaranda@usc.es (J.M.S.-P.); 2Health Research Institute of Santiago de Compostela (FIDIS), Santiago de Compostela University Clinical Hospital, University of Santiago de Compostela, Choupana Street, s/n, 15706 Santiago de Compostela, Spain; laurarojoalvarez@gmail.com; 3Department of Precision Medicine, University of Campania Luigi Vanvitelli, Via Abramo Lincoln, 5, 81100 Caserta, Italy; marina.didomenico@unicampania.it (M.D.D.); andrea.ballini@unifg.it (A.B.); 4Department of Clinical and Experimental Medicine, University of Foggia, Via Napoli, 20, 71122 Foggia, Italy; vitocarlo.caponio@unifg.it; 5Human Anatomy and Embryology Area, Department of Functional Biology and Health Sciences, University of Vigo, Lagoas-Marcosende, s/n, 36310 Vigo, Spain

**Keywords:** glioblastoma, DNA methylation, pyrosequencing, MGMT, survival

## Abstract

The O-6-methylguanine-DNA methyltransferase (MGMT) gene is a critical guardian of genomic integrity. MGMT methylation in diffuse gliomas serves as an important determinant of patients’ prognostic outcomes, more specifically in glioblastomas (GBMs). In GBMs, the absence of MGMT methylation, known as MGMT promoter unmethylation, often translates into a more challenging clinical scenario, tending to present resistance to chemotherapy and a worse prognosis. A pyrosequencing (PSQ) technique was used to analyze MGMT methylation status at different cut-offs (5%, 9%, and 11%) in a sample of 78 patients diagnosed with IDH-wildtype grade 4 GBM. A retrospective analysis was provided to collect clinicopathological and prognostic data. A statistical analysis was used to establish an association between methylation status and treatment response (TR) and disease-specific survival (DSS). The patients with methylated MGMT status experienced progressive disease rates of 84.6%, 80%, and 78.4% at the respective cut-offs of 5%, 9%, and 11%. The number was considerably higher when considering unmethylated patients, as all patients (100%), regardless of the cut-off, presented progressive disease. Regarding disease-specific survival (DSS), the Hazard Ratio (HR) was HR = 0.74 (0.45–1.24; *p* = 0.251); HR = 0.82 (0.51–1.33; *p* = 0.425); and HR = 0.79 (0.49–1.29; *p* = 0.350), respectively. Our study concludes that there is an association between MGMT unmethylation and worse TR and DSS. The 9% cut-off demonstrated a greater potential for patient survival as a function of time, which may shed light on the future need for standardization of MGMT methylation positivity parameters in PSQ.

## 1. Introduction

Diffuse gliomas (DGs) are a diverse group of brain tumors that arise from glial cells and are characterized by their aggressiveness and resistance to treatment, which causes high mortality rates and an urgent need for strategies that can improve the prognosis [1]. According to GLOBOCAN, in 2020, the official number of gliomas of all types was estimated at 298,000 new cases and 241,000 deaths, approximately [2].

The World Health Organization’s (WHO) latest classification of DGs offers a profound glimpse into the intricate tapestry of central nervous system (CNS) tumors with a cascade of molecular biomarker analyses, which expands to personalize the classification while also segmenting their subtypes from diagnosis to direct treatments and predicting the prognosis of each patient [3,4,5,6]. Particularly when it comes to distinguishing between astrocytomas and glioblastomas (GBMs), at the core of this differentiation lies a key piece of genetic code—the Isocitrate Dehydrogenase (IDH) gene [3,7].

The IDH gene works like a genetic marker and acts as a molecular compass, guiding clinicians and researchers through the labyrinth of glioma subtypes (Figure 1) [8]. GBMs, known for their aggressive and relentless nature, typically exhibit the IDH-wildtype status, signifying the absence of the characteristic mutation. In stark contrast, astrocytomas, often less aggressive, are marked by the presence of IDH mutations. This genetic divergence extends beyond mere nomenclature, profoundly impacting treatment options, prognosis, and patient outcomes [7,9,10].

The IDH gene plays a crucial role in cellular metabolism and genomic stability. Specifically, IDH enzymes participate in the tricarboxylic acid (TCA) cycle, where they catalyze the conversion of isocitrate to alpha-ketoglutarate. This enzymatic function is vital for energy production and the synthesis of important cellular building blocks. However, the significance of the IDH gene extends beyond metabolic pathways [11,12].

The acquisition of mutant IDH leads to a significant reprogramming of cellular metabolism, explaining the slow-growing nature of this disease and suggesting that targeting specific metabolic patterns of IDH mutants could be a valuable therapeutic approach [10]. Mutations in isocitrate IDH1 and, less commonly, IDH2 are defining features of most diffuse adult gliomas [9]. While most IDH-mutant DGs harbor the common arginine-to-histidine mutation at codon 132 (p.R132H), approximately 10% of cases exhibit non-canonical IDH mutations, characterized by distinct radiological and histological features and a potential association with cancer predisposition genes [10]. Mandatory immunohistochemical staining for the most prevalent mutant form of IDH1 (R132H) is recommended for diagnostic purposes in all DG samples. In cases where staining is negative, sequencing of IDH1 (codon 132) and IDH2 (codon 172) should be performed [8]. This distinction is crucial for an integrated diagnosis, particularly in patients with grade 2 and grade 3 DGs and those under 55 years of age with grade 4 tumors [10].

After analyzing the mutational status of the IDH 1/2 genes, it is also crucial to determine whether ATRX is retained or lost. In IDH-mutant tumors with retained ATRX, it is essential to assess whether 1p/19q codeletion is present. If 1p/19q is codeleted and TERT is also mutated, the classification as oligodendroglioma is unequivocal. If 1p/19q and CDKN2A/B are retained and the tumor lacks necrosis and/or microvascular proliferation (MVP), the classification is a grade 2 or 3 astrocytoma. In IDH-mutant tumors with lost ATRX, the presence or absence of a homozygous deletion of CDKN2A/B can be directly assessed. If CDKN2A/B is homozygously deleted, the classification becomes an astrocytoma grade 4. In cases where CDKN2A/B is retained, necrosis and/or MVP need to be evaluated further. An astrocytoma grade 2 or 3 is diagnosed if these features are absent, whereas the diagnosis becomes an astrocytoma grade 4 if they are present [3,4,5,6,9].

In IDH-wildtype tumors with retained ATRX, the presence of necrosis and/or MVP or TERT mutation, EGFR amplification and/or +7/−10, and H3.3G34 wildtype status warrant the classification as GBM. However, if H3.3G34 is mutated, the classification shifts to a diffuse hemispheric glioma. IDH-wildtype tumors with retained ATRX in the thalamus, brainstem, or spinal cord are considered diffuse midline gliomas. However, the most crucial aspect is to evaluate these tumors for histone H3 K27M mutations and loss of nuclear K27-trimethylated histone H3 (H3K27me3) to definitively identify H3 K27M-mutant diffuse midline gliomas [3,4,5,6,9].

The O-6-methylguanine-DNA methyltransferase (MGMT) gene is a critical guardian of genomic integrity. Its primary function is to repair DNA damage by encoding the MGMT protein and removing alkyl groups, such as methyl or alkylating agents, from the O-6 position of guanine bases [13]. In summary, MGMT effectively safeguards against numerous mutations triggered by base transitions resulting from the presence of methylated bases. This DNA repair mechanism is crucial in preventing mutations that can lead to cancer [14].

In the context of DGs and various other cancers, the MGMT gene takes center stage due to its role in chemotherapy resistance. When the MGMT gene is hypermethylated, its expression is silenced, rendering the tumor cells vulnerable to the effects of alkylating chemotherapy agents like temozolomide (TMZ) [13,14,15,16].

DNA methylation is an epigenetic phenomenon involving chemical modifications that instigate heritable genetic changes, yet it preserves the underlying DNA sequence intact. These modifications lead to epigenetic silencing and bear significant relevance to processes like cellular differentiation and gene expression regulation [15].

MGMT methylation in DGs serves as an important determinant of patients’ prognostic outcomes, more specifically in GBMs. In GBMs, the absence of MGMT methylation, known as MGMT promoter unmethylation, often translates into a more challenging clinical scenario, tending to present resistance to chemotherapy. The same occurs in astrocytomas, in which MGMT promoter methylation generally means a more favorable prognosis [17,18,19].

This epigenetic silencing of the MGMT gene increases the sensitivity of tumor cells to alkylating chemotherapeutic agents, leading to better treatment responses and potentially prolonged survival [18,19]. The intricate interplay between MGMT methylation status and prognosis underscores the critical role of epigenetics in defining the course of these aggressive CNS tumors, ultimately guiding therapeutic decisions and offering hope to both patients and clinicians [3,16,20,21,22,23,24,25].

Analyzing the methylation status of the MGMT promoter region has become a crucial diagnostic and prognostic tool in the management of gliomas and other malignancies [26,27,28,29,30,31]. Methylation-specific polymerase chain reaction (MSP) and bisulfite sequencing are commonly employed methods to assess MGMT promoter methylation. Pyrosequencing (PSQ) is a sequencing-by-synthesis approach that utilizes the real-time incorporation of nucleotides and monitors the enzymatic conversion of released pyrophosphate into a proportional light signal for quantitative analysis. Quantitative measurements are particularly relevant for DNA methylation analysis in diverse developmental and pathological settings. PSQ-based DNA methylation analysis combines a straightforward reaction protocol with reproducible and accurate measurements of methylation levels at multiple CpG sites within proximity, offering high quantitative resolution [32,33].

This study involved a retrospective analysis of medical records of patients diagnosed with GBM at the Santiago de Compostela University Clinical Hospital in Santiago de Compostela, Spain, during the period from January 2019 to November 2022 and aimed to compare the methylation results of MGMT using PSQ with the treatment response (TR) and disease-specific survival (DSS) of these patients.

## 2. Results

### 2.1. Sample Description

In all samples, the IDH status was analyzed via IHC, and only GBM cases were further included in this study (Figure 2). Of the 78 patients diagnosed with GBM, 57.7% (45) were male and 42.3% (33) were female. The average age of the patients at the time of diagnosis was 63.5 years old, with the minimum age being 16 and the maximum age being 86. A total of 38.5% (30) of patients were over 70 years old at the time of diagnosis. As of November 2023, of all patients diagnosed between January 2019 and November 2022, only 12.8% (10) were alive, while 87.2% (68) had died. Of the 10 living patients, 60% (6) have a good TR and a stable disease status, and 40% (4) currently have no TR and a progressive disease status. Of the 68 patients who died, 4.4% (3) died after relapses subsequent to a positive first TR (Table 1; Appendix A).

### 2.2. Association of Clinicopathological Parameters with MGMT Methylation Status

Among the 45 male patients, the average MGMT methylation was 20.13%, while in the 33 female patients, this average was higher, at 22.11%. Regarding age, in the 30 patients aged 70 years or older at the time of diagnosis, the average methylation rate was 22.10%, while in patients aged less than 70 years, it was 19.27%. Considering survival, among the 68 deceased patients, the average methylation rate was 19.09%, while in the 10 alive patients, this average was significantly higher, at 31.23% (Table 1).

### 2.3. MGMT Methylation Status

The average methylation, considering all 78 patients, was 20.6%, ranging from 1.5% to 80.5%. Regarding methylation levels in the CpG islands, the results of two examples are showed in Figure 3. The range varied between the extremes of 0% as the minimum and 100% as the maximum methylation rate.

#### 2.3.1. Cut-Off of 5% for Methylation Positivity

The methylation status of MGMT was 67.7% (52) methylated and 31.3% (26) unmethylated (Table 2). Considering TR, of the 26 unmethylated patients, 100% (26) presented progressive disease, while in the 52 methylated patients, this number dropped to 84.6% (44) (*p* < 0.001) (Figure 4A). Regarding DSS, only 7.7% (2) of the 26 unmethylated patients were alive after the follow-up, a number that increases considerably in the group of 52 methylated patients, reaching 15.4% (8). The Hazard Ratio (HR) for DSS was HR = 0.74 (0.45–1.24; *p* = 0.251) (Figure 5A).

#### 2.3.2. Cut-Off of 9% for Methylation Positivity

The methylation status of MGMT was 51.3% (40) methylated and 48.7% (38) unmethylated (Table 2). Considering TR, of the 38 unmethylated patients, 100% (38) presented progressive disease, while in the 40 methylated patients, this number dropped to 80% (32) (*p* < 0.001) (Figure 4B). Regarding DSS, only 5.3% (2) of the 38 unmethylated patients were alive after the follow-up, a number that increases considerably in the group of 40 methylated patients, reaching 20% (8). The HR for DSS was HR = 0.82 (0.51–1.33; *p* = 0.425) (Figure 5B).

#### 2.3.3. Cut-Off of 11% for Methylation Positivity

The methylation status of MGMT was 47.4% (37) methylated and 52.6% (41) unmethylated (Table 2). Considering TR, of the 41 unmethylated patients, 100% (41) presented progressive disease, while in the 37 methylated patients, this number dropped to 78.4% (29) (*p* < 0.001) (Figure 4C). Regarding DSS, only 2.4% of the 41 unmethylated patients were alive after the follow-up; this number increases considerably in the group of 37 methylated patients, reaching 24.3%. The HR for DSS was HR = 0.79 (0.49–1.29; *p* = 0.350) (Figure 5C).

## 3. Discussion

In general terms, the analysis of MGMT methylation is already a widely debated topic in the literature, emphasizing its role as an important factor to be analyzed in GBM, so much so that in the 2021 WHO standards, among the molecular cascade that dissects the diagnostic classification of CNS tumors, MGMT is already essential in cases of GBM as a predictive value for prognosis in these types of tumors and also for making therapeutic decisions [1].

In our study, the average survival of the patients was 16 months, a result similar to the majority of the studies in the literature, such as Ramos-Fresnedo et al.’s [11], in which the average was 13 months, and Witthayanuwat et al.’s, in which the average was 12 months [34]. However, it differs quite a lot from other studies, such as that by Dahlrot et al. [35], in which the average was 9 months, and that of Audureau et al. [36], in which it was 8 months.

Many factors can influence these changes in survival time, obviously including the individual clinicopathological characteristics of each patient, such as the ethnic–geographical limitations relating to each study and the types of therapeutic protocols employed, for example [37]. The year in which the studies were carried out could also have a direct influence since the classification of CNS tumors has undergone some changes over the last few years, as new molecular characteristics have been dissected and diagnostic subdivisions have been better defined and built over the years [1].

Many of the studies carried out before the last WHO resolution did not consider the current characteristics that our study considered, for example, the importance of analyzing the mutation in the IDH gene as a differentiating factor between grade 4 astrocytomas and GBMs. Previously, astrocytomas were classified only as grades 1, 2, and 3, while GBM was the name given to grade 4 of this type of tumor. Today, astrocytomas can also be classified as grade 4, differentiating themselves from GBMs according to the IDH profile, with wildtype status in GBM and mutated status in astrocytomas [1,10,17].

This could make a complete difference when it comes to survival profiles since grade 4 astrocytomas are tumors with better prognostic factors than GBMs [10,38,39,40]. In our study, for example, the aggressiveness of these types of tumors can be seen when we observe that, after the follow-up period up to November 2023, only 12.8% of the 78 patients were alive.

Our study used inclusion factors for very specific patients’ diagnostics so that we could maintain the fidelity of the results using the new WHO classification for CNS tumors. For this reason, there were few clinicopathological divergences to be debated; among these, we observed that there were no significant differences in the percentages of methylation related to gender and age. Considering survival status, we found a significantly higher average MGMT methylation in living patients compared to dead patients (31.23% and 19.09%, respectively), indicating a direct effect of MGMT methylation on the survival of these patients.

Another point of divergence that our study aimed to clarify was the cut-off to be used as a parameter for positive MGMT methylation. This is because, in general, there are many divergences regarding the methylation profile to be considered as a classification. For example, Barbagallo et al., Lattanzio et al., González-Jimenéz et al. and Gurriere et al. used a cut-off of 9% [41,42,43,44,45], while Quilien et al. and Havik et al. used a cut-off of 8% [32,46], Hsu et al. a cut-off of 5% [47], and Kristensen et al. and Dong Shen et al. a cut-off of 10% [48,49]. Many authors use sensitive analysis methods to define the cut-off, e.g., Receiver Operating Characteristic (ROC) analyses [27,50,51].

With the divergences surrounding the parameters of positivity for MGMT methylation through PSQ, there is an urgent need for greater definitions that can avoid distortions and contrasts between these analyses and their determined prognostic values, providing more precision in the association between this methylation status and the prognostic prediction of the patient at the time of diagnosis [52,53,54]. Therefore, our study analyzed the prognostic and survival values associated with this group of patients at different cut-offs. Three different cut-off points were used, two according to the literature at 5% and 9% [34,35,36,37,38,39] and a third at 11% [55]. The values of 8% and 10%, also quite common in the literature, were excluded due to the small difference in the number of patients between these groups and the 9% group.

In our study, when considering the MGMT methylation status at the cut-off, the difference between 9% and 11% was subtle. However, the comparison between the 5% group and the 9%/11% groups showed a robust difference, which could change the direction of treatment and the predictive value of prognosis. The methylation of the MGMT gene influenced TR because unmethylated patients always presented progressive disease/non-response to treatment, while when dealing with methylated patients, some of the patients presented stable disease/good response to treatment. The comparison between methylated and unmethylated patients was statistically significant, pointing to a greater chance of stable disease or good TR in patients where MGMT was methylated.

Our study also noticed an improvement in the percentage of patients corresponding to the increase in the percentage of cut-off used, pointing out an important association between the percentage of MGMT methylation and TR, demonstrating that quantitative, not just qualitative, methylation values should be considered to predict TR. In opposition to this, Hegi et al. found that there may be no additional survival benefit after a particular degree of methylation of the MGMT promoter region [56].

A large number of studies have reached a conclusion similar to ours regarding the unmethylation of MGMT with a worse TR [57,58], especially in cases of patients treated with TMZ. Malmström et al., Nagane et al., and Silber et al. demonstrated in their results that the MGMT promoter methylation status is an important factor in predicting the benefit of TMZ administration [59,60,61]. Mansouri et al. [14] concluded that MGMT expression can be induced in response to TMZ, and in patients over 65 years of age, MGMT unmethylation appears to have an even greater effect on TMZ resistance. In contrast, studies such as those by Gilbert et al. and Butler et al. did not demonstrate greater efficacy of TMZ regardless of methylation status [62,63].

Our study revealed a robust association between MGMT methylation and GBM patient DSS, with patients harboring unmethylated MGMT exhibiting a significantly poor survival rate. This association was evident when patients were stratified at the 9% methylation cut-off, at which patients with unmethylated MGMT had a Hazard Ratio of 0.82 (0.51–1.33; *p* = 0.425), indicating a lower likelihood of survival compared to those with methylated MGMT.

The survival benefit associated with a silenced MGMT gene was first described by Hegi et al. in 2005 [56]. Since then, along with TR, survival has been consistently associated with MGMT methylation status by many authors [64,65,66]. Caccese et al., for example, associated a higher risk of mortality with HR = 2.45 (1.98 to 3.05; *p* < 0.0001) in unmethylated patients, as did Mikkelsen et al., who concluded that methylated MGMT patients survived significantly longer than unmethylated ones (*p* = 0.048) [22,27]. This was questioned by Gibson et al., who demonstrated a non-linear relationship between MGMT promoter methylation and patient survival [65]. Butler et al. suggested that MGMT methylation should not be used as a prognostic biomarker alone but rather in conjunction with other molecular characteristics individual to each patient [63].

Our study’s biggest limiting factor is the fact that GBM is an extremely aggressive tumor with a low survival rate. In our sample size, only 10 patients survived after the follow-up, which, in absolute numbers, may cause a greater gap between the percentages presented in the results and MGMT methylation. This limitation was resolved through statistical analyses that enabled an exact measurement of the treatment effect and patient survival between subgroups as a function of time, thus discriminating more faithful values of the effects of MGMT methylation with a long-term patient follow-up.

## 4. Material and Methods

### 4.1. Patient Selection

The cohort included in this retrospective study comprises randomly selected patients diagnosed with GBM. These patients received treatment at the Santiago de Compostela University Clinical Hospital in Santiago de Compostela, Spain, and the clinical and pathological data were retrieved from the database of the Galician Health Service in Galicia, Spain. Inclusion criteria encompassed: (a) histopathological diagnoses of GBM grade 4; (b) patients with IDH gene status; (c) MGMT gene methylation analysis conducted using PSQ; and (d) patient follow-up data up to either the time of death or the last data collection month in November 2023 (at least 1 year of follow-up). Exclusion criteria comprised: (a) absence of IDH gene status; (b) absence of MGMT methylation analysis using PSQ; and (c) patients with insufficient follow-up data.

The clinical parameters under examination included TR and DSS. The follow-up period was calculated from the date of diagnosis to the date of cancer-related death or the date of the last follow-up. The study protocol was reviewed and approved by the Santiago–Lugo Research Ethics Committee (CEI-SL), with registration code 2023/157. All procedures performed, as they were studies involving human participants, were in accordance with the Declaration of Helsinki.

### 4.2. Immunohistochemistry (IHC)

The patients were analyzed based on their classification according to the subtype of DG, considering only IDH-mutated grade 4 diffuse astrocytomas and GBM (non-mutated IDH) cases. The IHC procedure began with the dewaxing and rehydration of 5 µm paraffin-embedded tissue sections using a suitable dewaxing solution and graded alcohols, ensuring complete removal of the paraffin wax. Antigen retrieval was then performed using EnVision™ FLEX target retrieval solution (Agilent, DAKO, Santa Clara, CA, USA) at pH 10 for 20 min at 95 °C. Slides were cooled and treated with EnVision™ FLEX peroxidase-blocking reagent solution (Agilent, DAKO, USA) for 5 min. Sections were incubated with anti-DH1 R132H (monoclonal antibody, 1:50 dilution; Dianova, Hamburg, Germany) and/or IDH2 GT673 (monoclonal antibody, 1:100 dilution; Thermo Fisher Scientific, Waltham, MA, USA) primary antibody in EnVision™ FLEX antibody diluent and incubated with the diluted antibody at room temperature for 20 min. Complete immunostaining was performed using the EnVision™ FLEX + Mouse LINKER/HRP (Agilent, DAKO, Santa Clara, CA, USA) technique following the manufacturer’s instructions. Nuclei were counterstained with hematoxylin to provide contrast, and slides were mounted using a suitable mounting medium.

#### IHC Evaluation

The staining evaluation of the histopathological slides was always carried out by an expert anatomopathologist, and positivity was considered whenever the staining was diffused throughout the entire area of the tumor, in approximately more than 50% of the cytoplasm.

### 4.3. Patient Grouping

The patients were analyzed based on grade and IDH 1/2 gene profile, as suggested by the WHO classification for DGs. Patients were also dichotomously classified as MGMT methylated or unmethylated. Furthermore, to investigate discrepancies in MGMT methylation positivity, patients were also categorized according to different cut-off values of 5%, 9%, and 11% [41,42,43,44,45,46].

### 4.4. Data Extraction

The assessed qualitative variables encompassed the following parameters: patient number references; hospital of reference; gender; age; WHO classification diagnostic; grade; date of diagnosis; date of death (in case of); survival status; duration of follow-up; response to treatment; tumor recurrence; the revised WHO classification of relapsed cases; date of recurrence; MGMT methylation percentages; and a range of methylation (Appendix A).

### 4.5. Analysis of MGMT Methylation

#### 4.5.1. PSQ

To determine the methylation status of the samples within the MGMT gene, we conducted a PSQ analysis. Initially, the samples were embedded in paraffin. Total DNA extraction was carried out and subsequently treated with bisulfite using the Epitect Fast DNA Bisulfite Kit (Qiagen^®^, Venlo, The Netherlands), following the manufacturer’s recommended protocol. Specifically, two 5 μm sections were taken from biopsy cases for this purpose. The samples’ concentrations were quantified using NanoDrop 2000 (Thermo Fisher Scientific, Waltham, MA, USA), with all samples exhibiting 260/280 values ranging from 1.7 to 2. Once transformed, the DNA underwent extraction, and the Therascreen MGMT Pyro kit (Qiagen^®^, Venlo, The Netherlands) was employed for PCR amplification and subsequent PSQ to detect methylated MGMT. All PCR reactions were conducted using the Agilent Technologies Surecycler 8800 (Agilent, DAKO, Santa Clara, CA, USA), while PSQ was performed using the Pyromark Q24 and Pyromark Q24 workstation (Qiagen^®^, Venlo, The Netherlands), with the Pyromark Q24 2.0.7 software used for result computation and analysis.

#### 4.5.2. Quantification of Methylation

For each sample, the average methylation levels of the four CpG islands were respectively calculated. A positive methylation status (indicating “methylated”) was assessed using different cut-offs, such as 5% [46], 9% [41,42,43,44,45] (discussed in the literature), and 11%, to investigate possible statistical divergences in the prognostic results. In cases where the result was inconclusive, individual assessments of the corresponding CpG island and pyrogram were conducted. The four CpG islands were in exon 1 of the human MGMT gene (genomic sequence on chromosome 10 from 131, 265, 519 to 131, 265, 537: CGACGCCCGCAGGTCCTCG).

### 4.6. Statistical Analysis

This study adhered to the guidelines outlined in the Reporting Recommendations for Tumor Marker Prognostic Studies (REMARK) [67]. The normality of variables was assessed using the Shapiro–Wilk test, revealing non-normal distributions. For dichotomous variables, the chi-square test was applied, specifically investigating the proportions of patients experiencing different treatment outcomes with diverse methylation statuses. Linear variables were analyzed using the Mann–Whitney or Kruskal–Wallis tests. DSS was evaluated by performing Kaplan–Meier analysis with log-rank tests in univariate analysis using SPSS 21.0, with *p*-values < 0.05 considered indicators of statistical significance.

## 5. Conclusions

Our study concludes that there is a strong association between MGMT gene unmethylation and worse TR and DSS. It is also possible to consider in our results a possible association between a higher percentage of methylation and a greater chance of good TR. The 9% cut-off demonstrated a greater potential for patient survival as a function of time, which may shed light on the future need for standardization of MGMT methylation positivity parameters in PSQ.

## Figures and Tables

**Figure 1 ijms-25-00612-f001:**
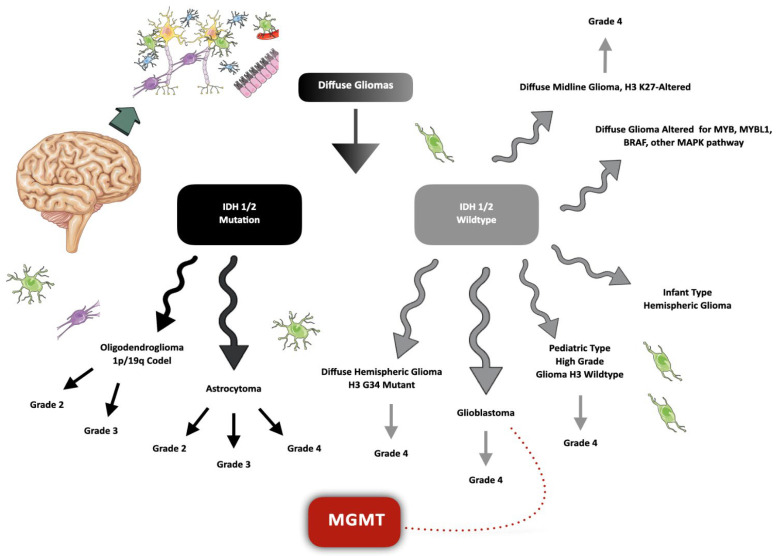
Latest classification of diffuse gliomas. A molecular pathway expands to personalize the classification of diffuse gliomas. An important step concerns the analysis of the mutation in the Isocitrate Dehydrogenase (IDH) 1 and 2 genes, which are responsible for the main segmentation in the current classification of this type of tumor. The O-6-methylguanine-DNA methyltransferase (MGMT) gene serves as an important determinant of patients’ prognostic outcomes, more specifically in IDH-wildtype grade 4 glioblastomas (GBMs), where the unmethylation tends to present resistance to chemotherapy and a worse prognosis.

**Figure 2 ijms-25-00612-f002:**
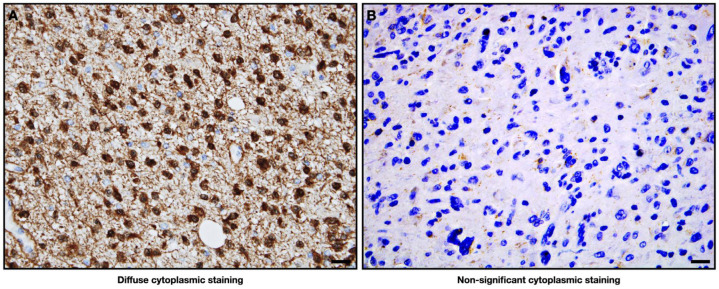
Representative IDH mutation by IDH protein expression analysis in immunohistochemistry. (**A**) Positivity is indicated by significant cytoplasmatic staining, which means an IDH gene mutation. Represented by a diagnosed case of IDH-mutated grade 4 astrocytoma; (**B**) the absence of staining is indicative of no expression of the protein and, consequently, of a wildtype status for the IDH gene. Represented by a diagnosed case of IDH-wildtype grade 4 GBM. Both cases included in this study are real; The antibodies used were anti-IDH1 R132H (monoclonal antibody, 1:50 dilution; Dianova, Hamburg, Germany) and/or IDH2 GT673 (monoclonal antibody, 1:100 dilution; Thermo Fisher Scientific, Waltham, MA, USA); 20× magnification.

**Figure 3 ijms-25-00612-f003:**
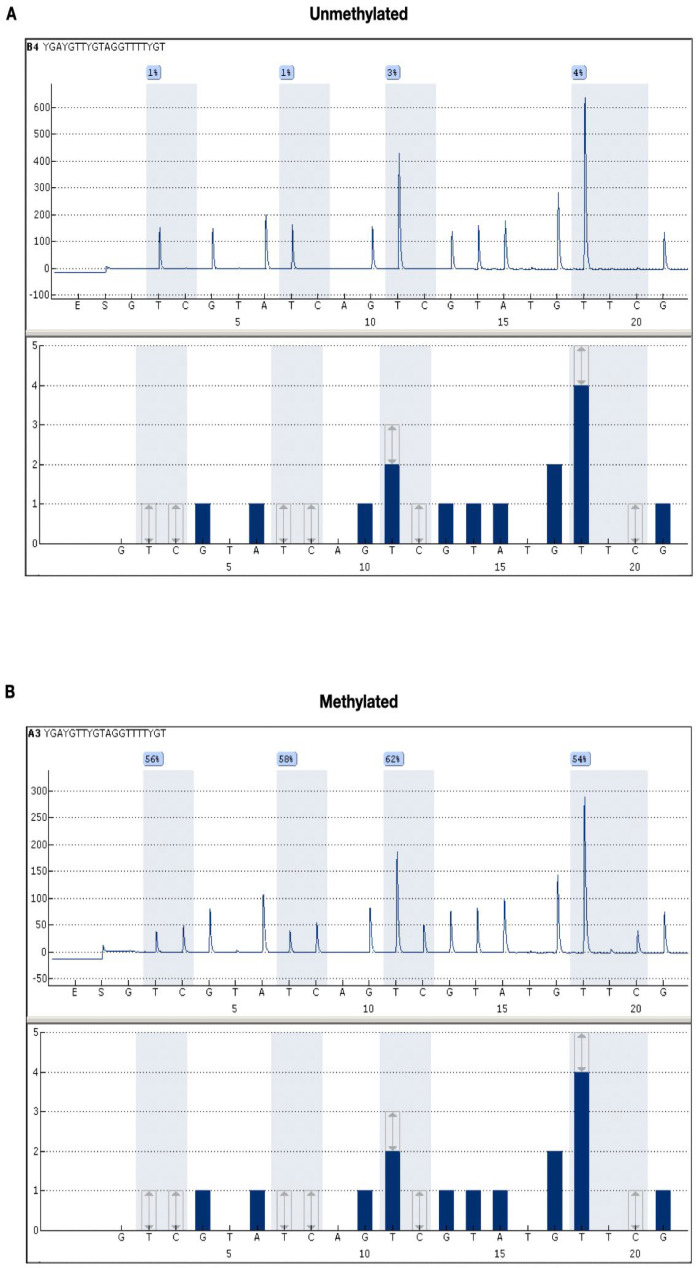
Representative pyrosequencing (PSQ) to determine MGMT methylation status. (**A**) The average methylation levels of the four CpG islands (island 1 = 1%; island 2 = 1%; island 3 = 3%; island 4 = 4%), with the representative average being 2.25%; (**B**) the average methylation levels of the four CpG islands (island 1 = 56%; island 2 = 58%; island 3 = 62%; island 4 = 54%), with a representative average of 57.5%. Therefore, regardless of the cut-off used, figure (**A**) represents an unmethylated case, and figure (**B**) represents a hypermethylated case. Both cases included in this study are real.

**Figure 4 ijms-25-00612-f004:**
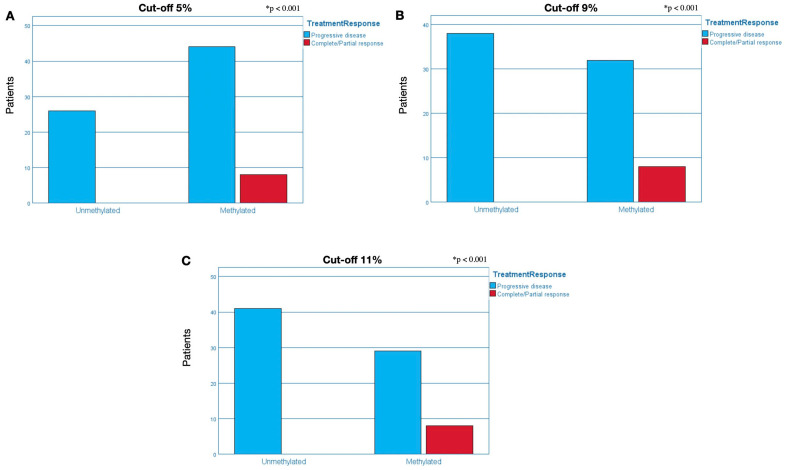
Analysis of response to treatment concerning MGMT methylation. The analyses were separated into two groups: unmethylated patients (right bars) and methylated patients (left bars). In the group composed of unmethylated patients, we can only observe a blue bar and the absence of a red bar, which is indicative that all patients (100%) in this group presented progressive disease/non-response to treatment, regardless of the cut-off. In the group of methylated patients, (**A**) 84.6%, (**B**) 80%, and (**C**) 78.4% presented a progressive disease/non-response to treatment (blue bar) at 5%, 9%, and 11% cut-off, respectively; and (**A**) 15.4%, (**B**) 20%, and (**C**) 21.6% presented stable disease/good response to treatment (red bar) at 5%, 9%, and 11% cut-off, respectively. * *p* < 0.001.

**Figure 5 ijms-25-00612-f005:**
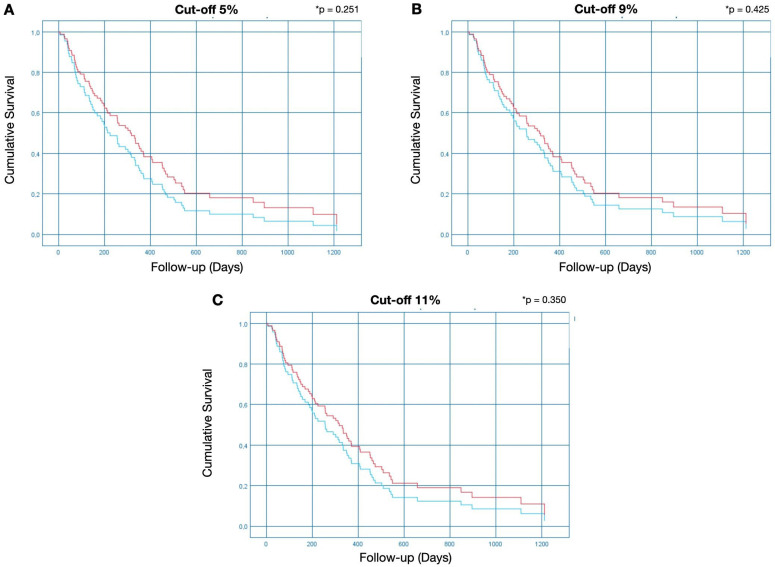
Disease-specific survival analysis. The Kaplan–Meier analysis represents each cut-off point used to indicate MGMT methylation positivity. The red line represents methylated patients, and the blue line represents unmethylated patients. The Y bar represents the event (in this case, the death of the patient due to the disease), and the X bar represents the follow-up time in days, from the moment of diagnosis until death or until the last day of follow-up (survival). The Hazard Ratio (HR) was (**A**) HR = 0.74 (0.45–1.24; * *p* = 0.251); (**B**) HR = 0.82 (0.51–1.33; * *p* = 0.425); and (**C**) HR = 0.79 (0.49–1.29; * *p* = 0.350) at cut-off points of 5%, 9%, and 11%, respectively.

**Table 1 ijms-25-00612-t001:** Qualitative variables in comparison with levels of methylation.

Qualitative Variables		Number of Patients	Average MGMT Methylation
Gender	Male	57.5% (45)	20.13%
	Female	42.3% (33)	22.19%
Age	<70 years old	61.5% (48)	19.27%
	≥70 years old	38.5% (30)	22.10%
Exitus	Alive	12.8% (10)	19.09%
	Dead	87.2% (68)	31.23%

**Table 2 ijms-25-00612-t002:** Segmentation of patients according to cut-off for classification of methylation status.

Cut-Off	Methylated	Unmethylated
5%	67.7% (52)	31.3% (26)
9%	51.3% (40)	48.7% (38)
11%	47.4% (37)	52.6% (41)

## Data Availability

Data are contained within the article and Appendix A.

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
