# Peer review of "Pyrosequencing Analysis of O-6-Methylguanine-DNA Methyltransferase Methylation at Different Cut-Offs of Positivity Associated with Treatment Response and Disease-Specific Survival in Isocitrate Dehydrogenase-Wildtype Grade 4 Glioblastoma"

_ijms, 2024, doi:10.3390/ijms25010612_

Round 1
Reviewer 1 Report
Comments and Suggestions for Authors
This manuscript investigates the prognostic significance of MGMT promoter methylation in GBM assessed through pyrosequencing and explores different cut-off values, potentially improving the methodology of MGMT promoter testing. Significant discoveries in the field of molecular biology/oncology of glioblastoma have occurred during the past decade, resulting in a better understanding of the process of gliomagenesis and more precise (molecular-based) classification. However, those discoveries gave rise to only minor improvements in clinical practice and the improvement in the survival of GBM patients.
Exploring the consequences of the molecular interplay between IDH1/2 mutations and MGMT promoter methylation stands out as a promising approach to defining novel therapeutic procedures. Given that, this manuscript presents data that is generally worth publishing. However, the editors should remember that many articles have already investigated the methodological aspects of MGMT promoter testing in gliomas/GBM.
I have some points that must be addressed before the article is suitable for publication:
- Although the introduction section makes reference to adequate literature and provides a satisfactory explanation of the main problem that this research tends to explore, I think it would be very beneficial if authors:
1.1. Provided a more thorough overview of IDH1/2 mutations' role in gliomagenesis and named the most common mutations correlated with gliomas. In addition, the sentence "However, the significance of the IDH gene extends beyond metabolic pathways" seems too general and redundant;
1.2. Presented a brief explanation of MSP and bisulfite-pyrosequencing methods, including their strengths and weaknesses presented in current literature, and emphasized the importance of defining the appropriate cut-off values. This is crucial since this represents one of the main goals and contributions of this research (which the authors addressed in the discussion).
2. The Material and Methods section is well-structured and provides sufficient information within most sections/paragraphs. Nevertheless, given the importance of IHC analyses:
2.1. I would ask the authors to provide more information regarding this procedure (antigen retrieval buffer, detection kit, etc.);
2.2. In addition, define cut-off values for discriminating IDH1/2 positive from the IDH1/2 negative IHC results (e.g., the percentage of stained cells).
3. This study presented convincing results that suggest a strong association of MGMT gene unmethylation with worse Treatment Response and Disease-Specific Survival. The study is generally well-planned and executed, and the results are presented with appropriate Figures and Tables. Statistical tests seem adequate and can be found within similar studies/articles (e.g., survival analysis, treatment response analysis). Here are some points that should be addressed within the Results section:
3.1. Introduce cut-off values below Figure 2 and check for the appropriate brightness setting of the B section of this Figure;
3.2. Please revise Figure 3 since differences between the (A) and (B) charts do not seem obvious (this may be due to the low resolution of Figure 3).
The Discussion section provides a solid analysis of the presented results with an overview of related and current data in the literature.In the final section, the authors concluded that their research found a strong association of MGMT gene unmethylation with worse TR and DSS. They also summarized that the 9% cut-off demonstrated a greater potential for patient survival as a function of time.
Overall, this manuscript presents valuable information worth publishing after revising the abovementioned issues.
Comments on the Quality of English LanguageMinor editing of English language required.
Author Response
- Firstly, we would like to thank each comment made by each reviewer, as each one of them seemed extremely pertinent and correct, with a clear intention of making improvements to our study through suggestions and/or corrections. All comments were taken into consideration and all changes and additions were made to bring our study into line with what was necessary for publication in IJMS.
This manuscript investigates the prognostic significance of MGMT promoter methylation in GBM assessed through pyrosequencing and explores different cut-off values, potentially improving the methodology of MGMT promoter testing. Significant discoveries in the field of molecular biology/oncology of glioblastoma have occurred during the past decade, resulting in a better understanding of the process of gliomagenesis and more precise (molecular-based) classification. However, those discoveries gave rise to only minor improvements in clinical practice and the improvement in the survival of GBM patients.
Exploring the consequences of the molecular interplay between IDH1/2 mutations and MGMT promoter methylation stands out as a promising approach to defining novel therapeutic procedures. Given that, this manuscript presents data that is generally worth publishing. However, the editors should remember that many articles have already investigated the methodological aspects of MGMT promoter testing in gliomas/GBM.
- We thank you for your insight and for considering the relevance of our study. We agree that there is a lot published in the literature, but there is a constant debate to be carried out, especially due to the lack of well-defined parameters for evaluating the methylation of this gene.
I have some points that must be addressed before the article is suitable for publication:
- Although the introduction section makes reference to adequate literature and provides a satisfactory explanation of the main problem that this research tends to explore, I think it would be very beneficial if authors:
1.1. Provided a more thorough overview of IDH1/2 mutations' role in gliomagenesis and named the most common mutations correlated with gliomas. In addition, the sentence "However, the significance of the IDH gene extends beyond metabolic pathways" seems too general and redundant;
- We fully agree that the indicated sentence was general and redundant. For this reason, we dissected the subject into 3 new paragraphs added in the introduction, to elucidate to the future reader of the article the importance of identifying IDH status in patients with diffuse gliomas.
- The following paragraphs have been added:
"The acquisition of mutant IDH leads to a significant reprogramming of cellular metabolism, explaining the slow-growing nature of this disease and suggesting that targeting specific metabolic patterns of IDH mutants could be a valuable therapeutic approach [10]. Mutations in isocitrate IDH1 and, less commonly, IDH2 are defining features of most diffuse adult gliomas [9]. While most IDH-mutant DG harbors the common arginine-to-histidine mutation at codon 132 (p.R132H), approximately 10% of cases exhibit non-canonical IDH mutations, characterized by distinct radiological and histological features and potential association with cancer predisposition genes [10]. Mandatory immunohistochemical staining for the most prevalent mutant form of IDH1 (R132H) is recommended for diagnostic purposes in all DG samples. In cases where staining is negative, sequencing of IDH1 (codon 132) and IDH2 (codon 172) should be performed [8]. This distinction is crucial for an integrated diagnosis, particularly in patients with grade 2 and grade 3 DG and those under 55 years of age with grade 4 tumors [10].
After analyzing the mutational status of the IDH 1/2 genes, it is also crucial to determine whether ATRX is retained or lost. In IDH-mutant tumors with retained ATRX, it is essential to assess whether 1p/19q codeletion is present. If 1p/19q is codeleted and TERT is also mutated, the classification is unequivocally oligodendroglioma. If 1p/19q and CDKN2A/B are retained, and the tumor lacks necrosis and/or microvascular proliferation (MVP), the classification is a grade 2 or 3 astrocytoma. In IDH-mutant tumors with lost ATRX, the presence or absence of homozygous deletion of CDKN2A/B can be directly assessed. If CDKN2A/B is homozygously deleted, the classification becomes an astrocytoma grade 4. In cases where CDKN2A/B is retained, necrosis and/or MVP need to be evaluated further. An astrocytoma grade 2 or 3 is diagnosed if these features are absent, whereas the diagnosis becomes an astrocytoma grade 4 if they are present [3-6, 9].
In IDH-wildtype tumors with retained ATRX, the presence of necrosis and/or MVP or TERT mutation, EGFR amplification and/or +7/-10, and H3.3G34 wildtype warrants the classification of GBM. However, if H3.3G34 is mutated, the classification shifts to a diffuse hemispheric glioma. IDH-wildtype tumors with retained ATRX in the thalamus, brainstem, or spinal cord are considered diffuse midline gliomas. However, the most crucial aspect is to evaluate these tumors for histone H3 K27M mutations and loss of nuclear K27-trimethylated histone H3 (H3K27me3) to definitively identify H3 K27M-mutant diffuse midline gliomas [3-6, 9]."
1.2. Presented a brief explanation of MSP and bisulfite-pyrosequencing methods, including their strengths and weaknesses presented in current literature, and emphasized the importance of defining the appropriate cut-off values. This is crucial since this represents one of the main goals and contributions of this research (which the authors addressed in the discussion).
- We agree that, because it is a technique, one of the differences of the study, this topic should be better addressed in our introduction. Therefore, we considerably expanded a paragraph in the introduction that addressed the topic, and information about pyrosequencing was added.
- The paragraph in question went like this:
"Analyzing the methylation status of the MGMT promoter region has become a crucial diagnostic and prognostic tool in the management of gliomas and other malignancies [26-31]. Methylation-specific polymerase chain reaction (MSP) and bisulfite sequencing are commonly employed methods to assess MGMT promoter methylation. Pyrosequencing (PSQ) is a sequencing-by-synthesis approach that utilizes the real-time incorporation of nucleotides and monitors the enzymatic conversion of released pyrophosphate into a proportional light signal for quantitative analysis. Quantitative measurements are particularly relevant for DNA methylation analysis in diverse developmental and pathological settings. PSQ-based DNA methylation analysis combines a straightforward reaction protocol with reproducible and accurate measurements of the methylation level at multiple CpG sites within proximity, offering high quantitative resolution [32, 33]."
2. The Material and Methods section is well-structured and provides sufficient information within most sections/paragraphs. Nevertheless, given the importance of IHC analyses:
2.1. I would ask the authors to provide more information regarding this procedure (antigen retrieval buffer, detection kit, etc.);
- To better understand each step of our technique, as well as the products used in our IHC step by step, we describe in detail how the technique was performed, through topic 2.2 of the article.
"2.2. Immunohistochemistry (IHC)
The patients were analyzed based on their classification according to the subtype of DG, considering only diffuse astrocytomas IDH-mutated grade 4 and GBM (non-mutated IDH) cases. The IHC procedure began with dewaxing and rehydration of 5-µm paraffin-embedded tissue sections using a suitable dewaxing solution and graded alcohols, ensuring complete removal of the paraffin wax. Antigen retrieval was then performed using EnVision™ FLEX target retrieval solution (Agilent, DAKO, USA) at pH10 for 20 minutes at 95°C. Slides were cooled and treated with EnVision™ FLEX peroxidase-blocking reagent solution (Agilent, DAKO, USA) for 5 minutes. Sections were incubated with anti-DH1 R132H (monoclonal antibody, 1:50 dilution; Dianova, Germany) and/or IDH2 GT673 (monoclonal antibody, 1:100 dilution; Thermo Fisher Scientific, USA) primary antibody in EnVision™ FLEX antibody diluent and incubated the sections with the diluted antibody at room temperature for 20 minutes. Complete immunostaining was performed using the EnVision™ FLEX + Mouse LINKER/HRP (Agilent, DAKO, USA) technique following the manufacturer's instructions. Nuclei were counterstained with hematoxylin to provide contrast and slides were mounted using a suitable mounting medium."
2.2. In addition, define cut-off values for discriminating IDH1/2 positive from the IDH1/2 negative IHC results (e.g., the percentage of stained cells).
- We understand that assuming that readers already know these parameters is a mistake, and therefore, we describe in a new topic added to our materials and methods, how the evaluation was carried out.
"2.2.1 IHC Evaluation
The staining evaluation of the histopathological slides was always carried out by an expert anatomopathologist, and positivity was considered whenever the staining was diffused throughout the entire area of the tumor, in approximately more than 50% of the cytoplasm."
3. This study presented convincing results that suggest a strong association of MGMT gene unmethylation with worse Treatment Response and Disease-Specific Survival. The study is generally well-planned and executed, and the results are presented with appropriate Figures and Tables. Statistical tests seem adequate and can be found within similar studies/articles (e.g., survival analysis, treatment response analysis). Here are some points that should be addressed within the Results section:
3.1. Introduce cut-off values below Figure 2 and check for the appropriate brightness setting of the B section of this Figure;
- We agree that the suggestion provides greater clarity in visualizing the results, and therefore, we have clarified part B of the figure. We also added a legend at the bottom of the figure, referring to positivity due to diffuse marking. We did not put exact cut-offs in the image, because this information has already been added in the new materials and methods topic.
3.2. Please revise Figure 3 since differences between the (A) and (B) charts do not seem obvious (this may be due to the low resolution of Figure 3).
- We try to improve the quality of the images and put them in high resolution, not just this one, but all the others. We also describe which image represents methylated and unmethylated at the top of the figure. And in each of the others, we also made improvements to make it easier for the reader to better understand our results.
The Discussion section provides a solid analysis of the presented results with an overview of related and current data in the literature. In the final section, the authors concluded that their research found a strong association of MGMT gene unmethylation with worse TR and DSS. They also summarized that the 9% cut-off demonstrated a greater potential for patient survival as a function of time.
Overall, this manuscript presents valuable information worth publishing after revising the abovementioned issues.
- We greatly appreciate each comment and suggestion, it was very valuable, and we certainly agree that each of the changes was necessary to increase the quality of the study. We also state that the English of every article has been revised, and improvements made. Thanks!

Reviewer 2 Report
Comments and Suggestions for Authors
In this manuscript the authors describe interesting results demonstrating an association of MGMT unmethylation with worse treatment response and disease-specific survival in patients with Glioblastoma IDH-Wildtype. These data underline the need for standardization of MGMT methylation quantification, which could represent a prognostic tool in the management of cancers, ultimately guiding therapeutic decisions and offering hope to both patients and clinicians
The introduction is well written, but I have some observations regarding the other sections:
1. It is necessary to specify which are the 4 CpG sites analyzed.
2. The legend in figure 2 should report the antibodies used and be clearer for readers.
3. The quality of figures 3 and 4 is poor, and the legends needs to be more detailed.
4. In Figure 4 it is not clear what the red bar represents.
5. La legenda della figura 5 non è chiara. The Y and X axes represent the death events and time of follow-up (survival) (what are the units?) or the results of Kaplan-Meier analysis of these parameters.
6. Statistical significance is not always reported in the results and is never included in the figures and tables.
7. In the discussion, the results are repeated, citing figures and tables again. It would be more useful to comment on them to make their meaning clearer.
Comments on the Quality of English LanguageSome editing for English language is required throughout the manuscript.
Author Response
In this manuscript, the authors describe interesting results demonstrating an association of MGMT unmethylation with worse treatment response and disease-specific survival in patients with Glioblastoma IDH-Wildtype. These data underline the need for standardization of MGMT methylation quantification, which could represent a prognostic tool in the management of cancers, ultimately guiding therapeutic decisions and offering hope to both patients and clinicians.
- First, we would like to thank each reviewer's comment, as each one seemed extremely pertinent and correct. All comments were taken into account and all changes and additions were made to make the improvements suggested by them.
The introduction is well written, but I have some observations regarding the other sections:
- It is necessary to specify which are the 4 CpG sites analyzed.
- We agree that this information was missing, and that is why we added the following information to our materials and methods, in topic 2.5.2.:
"The four CpG sites was in exon 1 of the human MGMT gene (genomic sequence on chromosome 10 from 131,265,519 to 131,265,537: CGACGCCCGCAGGTCCTCG)."
2. The legend in figure 2 should report the antibodies used and be clearer for readers.
- We agree that for greater clarity, the information should also be present in the caption, therefore, the following information was added:
"The antibodies used was anti-IDH1 R132H (monoclonal antibody, 1:50 dilution; Dianova, Germany) and/or IDH2 GT673 (monoclonal antibody, 1:100 dilution; Thermo Fisher Scientific, USA)."
3. The quality of figures 3 and 4 is poor, and the legends needs to be more detailed.
- We thank you for the comment, and we assure you that we have arranged to improve the quality of the images, and put them in high-resolution, not only this one, but also all the others, so that it would be more clear for the reader.
- Furthermore, in figure 3, we dissect in the caption what the figure corresponds to, through the following added information:
"Representative pyrosequencing (PSQ) to determine MGMT methylation status. (A) Demethylation is indicated by the average methylation levels of the four CpG islands (island 1 = 1%; island 2 = 1%; island 3 = 3%; island 4 = 4%) with the representative average being 2.25% methylation; (B) Methylation is indicated by the average methylation levels of the four CpG islands island 1 = 56%; island 2 = 58%; island 3 = 62%; island 4 = 54%) with a representative average of 57.5% methylation. Therefore, regardless of the cut-offs used, figure A represents an unmethylated case; and figure B a hypermethylated case. *Both are real cases included in the study."
- Concerning figure 4, we also changed the caption, as requested, and now the text has greater detail so that the reader can make a better reading of the results:
"Figure 4. Analysis of response to treatment concerning MGMT methylation. The analyses were separated into two groups, unmethylated patients (right bars) and methylated patients (left bars). In the group indicated by unmethylated patients, we can only observe a blue bar and the absence of a red bar, which is indicative that all patients (100%) in this group presented progressive disease/non-response to treatment, regardless of the cut-off. In the group of methylated patients (A) 84.6%, (B) 80%, and (C) 78.4% presented a progressive disease/non-response to treatment (blue bar), respectively, in 5%, 9%, and 11 % cut-off; and (A) 15.4%, (B) 20% and (C) 21.6%, presented stable disease/good response to treatment (red bar), respectively, at 5%, 9% and 11% cut-off. *p < 0.001"
4. In Figure 4 it is not clear what the red bar represents.
- We agree, and repeat the information from the previous suggestion, as the entire caption has been modified for greater understanding on the part of the reader.
"Figure 4. Analysis of response to treatment concerning MGMT methylation. The analyses were separated into two groups, unmethylated patients (right bars) and methylated patients (left bars). In the group indicated by unmethylated patients, we can only observe a blue bar and the absence of a red bar, which is indicative that all patients (100%) in this group presented progressive disease/non-response to treatment, regardless of the cut-off. In the group of methylated patients (A) 84.6%, (B) 80%, and (C) 78.4% presented a progressive disease/non-response to treatment (blue bar), respectively, in 5%, 9%, and 11 % cut-off; and (A) 15.4%, (B) 20% and (C) 21.6%, presented stable disease/good response to treatment (red bar), respectively, at 5%, 9% and 11% cut-off. *p < 0.001"
5. La legenda della figura 5 non è chiara. The Y and X axes represent the death events and time of follow-up (survival) (what are the units?) or the results of Kaplan-Meier analysis of these parameters.
- Figure 5, like all the others, was modified for greater clarity and understanding. As well as, following this suggestion, we changed the legend, so that the results can be viewed carefully and in greater detail:
"Figure 5. Disease-specific Survival Analysis. The Kaplan-Meier represents each cutoff point used to indicate MGMT methylation positivity; The red line represents methylated patients; The blue line represents unmethylated patients. The Y bar represents the event, in this case, the death of the patient due to the disease, and the X bar represents the follow-up time in days, from the moment of diagnosis until death or until the last day of follow-up (survival). The Hazard Ratio (HR) was (A) HR = 0.74 (0.45-1.24; p = 0.251); (B) HR = 0.82 (0.51-1.33; p = 0.425); (C) HR = 0.79 (0.49-1.29; p = 0.350), respectively at cutoff points of 5%, 9% and 11%."
6. Statistical significance is not always reported in the results and is never included in the figures and tables.
- We greatly appreciate the suggestion, and inform you that we have changed all the figures, adding significant representative significance to each of them.
7. In the discussion, the results are repeated, citing figures and tables again. It would be more useful to comment on them to make their meaning clearer.
- When rereading the discussion topic, we understood that the results were repeated in several parts of the text. This information was changed, where instead of demonstrating the numerical results, we inserted in text form what each result represents. We agree that this way, the discussion becomes more fluid and clear for the reader.
- We greatly appreciate each comment and suggestion, it was very valuable, and we certainly agree that each of the changes was necessary to increase the quality of the study. We also state that the English of every article has been revised, and improvements made. Thanks!
